# The Prognostic and Predictive Role of the Neutrophil-to-Lymphocyte Ratio (NLR), Platelet-to-Lymphocyte Ratio (PLR), and Lymphocyte-to-Monocyte Ratio (LMR) as Biomarkers in Resected Pancreatic Cancer

**DOI:** 10.3390/jcm12051989

**Published:** 2023-03-02

**Authors:** Sarah Maloney, Nick Pavlakis, Malinda Itchins, Jennifer Arena, Anubhav Mittal, Amanda Hudson, Emily Colvin, Sumit Sahni, Connie Diakos, David Chan, Anthony J. Gill, Jaswinder Samra, Stephen J. Clarke

**Affiliations:** 1Faculty of Medicine and Health Sciences, Northern Clinical School, The University of Sydney, Sydney, NSW 2065, Australia; 2Bill Walsh Translational Cancer Research Laboratory, Kolling Institute, The University of Sydney, Sydney, NSW 2065, Australia; 3Department of Medical Oncology, Royal North Shore Hospital, St. Leonards, NSW 2065, Australia; 4Upper Gastrointestinal Surgical Unit, Royal North Shore Hospital, St. Leonards, NSW 2065, Australia; 5Cancer Diagnosis and Pathology Group, Kolling Institute, The University of Sydney, Sydney, NSW 2065, Australia; 6NSW Health Pathology, Department of Anatomical Pathology, Royal North Shore Hospital, St. Leonards, NSW 2065, Australia

**Keywords:** inflammatory, immune, biomarkers, pancreatic cancer, predictive, prognostic

## Abstract

Pancreatic cancer has poor survival despite modern-day advances in its management. At present, there are no available biomarkers that can predict chemotherapy response or help inform prognosis. In more recent years, there has been increased interest in potential inflammatory biomarkers, with studies revealing a worse prognosis of patients with a higher neutrophil-to-lymphocyte ratio in a range of tumour types. Our aim was to assess the role of three inflammatory biomarkers in peripheral blood in predicting chemotherapy response in patients with earlier disease treated with neoadjuvant chemotherapy and as a prognostic marker in all patients that underwent surgery for pancreatic cancer. Using retrospective records, we discovered that patients with a higher neutrophil-to-lymphocyte ratio (>5) at the time of diagnosis had worse median overall survival than those with ratios ≤5 at 13 and 32.4 months (*p* = 0.001, HR 2.43), respectively. We were able to appreciate a correlation between a higher platelet-to-lymphocyte ratio and increased residual tumour in the histopathological specimen in patients receiving neoadjuvant chemotherapy; however, the association was weak (*p* = 0.03, coefficient 0.21). Due to the dynamic relationship between the immune system and pancreatic cancer, it is unsurprising that immune markers may be useful as potential biomarkers; however, larger prospective studies are needed to validate these findings.

## 1. Introduction

Pancreatic cancer has one of the highest mortality rates of any solid organ malignancy [1]. Most patients present with advanced disease, owing to non-specific symptoms in earlier stages. For those with locally advanced disease, the introduction of neoadjuvant chemotherapy has afforded select patients the opportunity to undergo curative surgery. Despite this, a high proportion of patients that undergo resection relapse early and have poor overall survival [2]. 

The challenge with more aggressive treatment is that it is often difficult to predict which patients will respond to chemotherapy and derive a meaningful survival benefit. There is a lack of clinically relevant, predictive, and prognostic biomarkers in this devastating disease. At present, both upfront radiological staging and resectability status are used to assist with prognostication; however, biological markers in this space are lacking. Cancer antigen 19.9 (CA 19.9) is currently used as a marker to assess response to therapy; however, its use as a prognostic biomarker is not validated. 

The role of circulating inflammatory cells, including neutrophils as prognostic markers, is well established in a range of different tumour types [3,4,5]. Several useful measures have emerged in recent years in patients with advanced disease, including the neutrophil-to-lymphocyte ratio (NLR), platelet-to-lymphocyte ratio (PLR), and lymphocyte-to-monocyte ratio (LMR), with higher numbers associated with worse prognosis and a reduction in overall survival [5]. Neutrophils in particular are known to produce reactive oxygen species, a key trigger in DNA damage and a hallmark in the development and progression of cancer [6]. 

In pancreatic cancer, there have been many mechanisms posited associating a higher number of neutrophils with more aggressive disease. One of these involves tumour secretion of chemokines, including CXCR2. It has been demonstrated that higher levels of CXCR2 expression are associated with an increase in tumour size in pancreatic adenocarcinoma and a worse prognosis in a range of cancer types [7,8]. Other in vitro studies (using transwell assays) have demonstrated that neutrophils promote the migration and invasion potential of pancreatic cancer cells [8].

The utility of these inflammatory markers in early disease is less established, with smaller tumours not generating as significant an immune response as more advanced cancers that have undergone haematogenous spread [9,10]. 

C-reactive protein (CRP) is an acute phase reactant that is produced by pro-inflammatory cytokines (including IL-6) [11]. It is a key marker of inflammation and is easily tested in routine bloods. Albumin is another protein that has several important functions, including the regulation of oncotic pressures in the blood vessels. Low levels can be associated with increased losses (gastrointestinal or renal), as a marker of malnutrition, or as a negative acute phase reactant in inflammation [12]. The latter two commonly occur in patients diagnosed with cancer. The modified Glasgow performance scale (mGPS), which incorporates both albumin and CRP levels, is becoming increasingly used as a prognostic marker in gastrointestinal cancers. Patients with lower levels of albumin and higher levels of CRP have worse outcomes [13]. 

There have been minimal clinical studies assessing these inflammatory tests as markers to predict response to chemotherapy in patients with pancreatic cancer. In the modern-day treatment of pancreatic cancer, upfront surgery has become the exception rather than the rule, with the majority of patients now undergoing neoadjuvant chemotherapy to downstage their disease. 

Tumour regression is assessed in a histopathological specimen from patients that have undergone neoadjuvant chemotherapy. It is an assessment of the proportion of viable tumour cells remaining in the surgical specimen in relation to the initial tumour volume [14]. As the initial volume is unknown, it is an estimate and takes into consideration the degree of fibrosis post-neoadjuvant treatment [15]. At present, many tumour regression scores are used, with the most common being the College of American Pathologists (CAP) score and Evans criteria. However, there exists significant interobserver variability using these tools [16]. Moreover, while it allows for an assessment of the chemosensitivity of the tumour, its use as a prognostic marker has not been established [15]. 

That there is a clear need for validated prognostic and biomarkers is indisputable. The utilisation of inflammatory blood tests, such as the NLR, PLR, and LMR, as biomarkers have widespread appeal. They are routinely performed on all patients at diagnosis, are minimally invasive, and are cost-effective. 

We hypothesise that due to the active role the immune system has in the development of pancreatic cancer, these inflammatory markers can be used as prognostic and predictive biomarkers in response to chemotherapy in early pancreatic cancer. 

The aims of this study were (1) to determine if inflammatory markers, including the NLR, PLR, and LMR, as well as other clinical markers, could predict tumour viability at the time of surgery in patients treated with neoadjuvant chemotherapy at our high-volume pancreatic cancer centre and (2) to assess if these markers were useful in prognostication for recurrence-free and overall survival in pancreatic cancer patients that underwent surgical resection. 

## 2. Materials and Methods

### 2.1. Patient Population and Clinical Characteristics

Patients that were treated at one large tertiary centre from 2014 to 2020 were identified from prospective clinical databases and medical records. Patients were included if they underwent surgery for localised pancreatic cancer and were identified as either upfront resectable, borderline resectable, or locally advanced (Table 1) [17]. Patients that were diagnosed with metastatic pancreatic cancer or did not proceed to surgical resection were excluded. 

All surgeries were performed by two specialised pancreatic cancer surgeons (JS and AM). Patient characteristics including age, sex, Eastern Co-operative Oncology Group (ECOG) performance status, and Charleson comorbidity index were collected (Table 1) [18,19]. Clinical symptoms, including weight loss or back pain, were also recorded. Baseline blood tests included inflammatory markers such as the neutrophil-to-lymphocyte ratio (NLR), platelet-to-lymphocyte ratio (PLR), lymphocyte-to-monocyte ratio (LMR), and c-reactive protein (CRP); albumin and bilirubin were also collected. These values were analysed as dichotomous variables, and cut-offs were chosen from previously published values and were NLR > 5, PLR > 150, LMR > 3, CA 19–9 > 1000 [4,9,20,21]. C-reactive protein (10 mg/L) and albumin (<35 g/L) were combined to generate a modified Glasgow performance score (mGPS), with a higher CRP (>10 mg/L) and lower albumin (<35 g/L) indicating a worse prognosis. Bilirubin (>20 µmol/L) was also measured as a dichotomous variable, and the cut-off was based on the testing laboratory’s standard reference range.

Blood tests were taken at the time of staging laparoscopy and prior to chemotherapy in patients that received neoadjuvant chemotherapy (NAC). Baseline resectability and stage were determined via a computerised tomography (CT) scan. Upfront treatment with either NAC or upfront surgery was recorded (Figure 1, Table 1).

Tumour regression was assessed only on the histopathological specimens of patients that received NAC. It was a marker of viable tumour remaining based on an estimate of initial tumour volume. This was assessed as a percentage and measured as a continuous variable. No scoring system was used due to the lack of consensus on the tumour regression grading system in pancreatic cancer. 

The histopathological stage was based on surgical histology using the AJCC TNM 8th edition [22]. Clinical factors, including recurrence-free (RFS) and overall survival (OS), were analysed. 

Patients were discussed at a multidisciplinary team meeting, and the decision to perform NAC versus upfront surgery was determined based on time of diagnosis and resectability (due to the success of NAC in the locally advanced setting, it became the standard of care for patients with upfront and borderline resectable disease from 2016 onwards in our centre). 

### 2.2. Statistical Analysis

Predictive markers were analysed using Spearman’s rank correlation with tumour viability. Prognostic markers were assessed using Kaplan–Meier and Cox proportional hazards using both univariate and multivariate analyses. Statistical analysis was performed using SPSS (IBM Corp, Armonk, NY, USA). *p*-values < 0.05 were considered statistically significant. 

### 2.3. Institutional Ethics

Institutional ethics approval for this study was obtained (HREC/16/HAWKE/105).

## 3. Results

One hundred and ninety-six patients were included in the analysis, and baseline characteristics are presented in Table 1. Of these, 110 received neoadjuvant chemotherapy (NAC) and 86 patients underwent upfront surgery (UFS, Figure 1). The majority of patients in the NAC cohort received either gemcitabine plus abraxane or FOLFIRINOX (5- fluorouracil, irinotecan, oxaliplatin) as per physician discretion. 

The majority (159/196) of patients had early disease (upfront resectable or borderline resectable disease) at diagnosis.

### 3.1. Predictive Biomarkers

To identify the presence of biomarkers that may predict chemotherapy response, baseline clinical characteristics and blood tests were compared to the residual tumour (tumour viability) at the time of surgery for the 110 patients in the NAC cohort (Figure 1). The patients who had upfront surgery were excluded. Of these, the platelet-to-lymphocyte ratio (PLR) correlated to tumour viability with a platelet-to-lymphocyte ratio > 150 predicting higher tumour viability and hence poorer chemotherapy response (*p* = 0.03, coefficient 0.21) at the time of surgery (Table 2). The neutrophil-to-lymphocyte ratio (NLR), lymphocyte-to-monocyte ratio (LMR), and modified Glasgow performance scale (mGPS) did not have any correlation with tumour viability.

### 3.2. Prognostic Biomarkers

To assess if baseline clinical inflammatory markers were useful as prognostic biomarkers, all 196 patients were included in the analysis, regardless of whether they received NAC or upfront surgery (Figure 1). Both univariate and multivariate analyses were performed to assess the role of these biomarkers in prognosis both in recurrence-free (RFS, Table 3) and overall survival (OS, Table 4). 

#### 3.2.1. Recurrence-Free Survival

Of these factors, the surgical stage was a significant prognostic marker in both univariate (*p* = 0.001, HR 1.36) and multivariate analyses (*p* = 0.003, HR 1.66, Table 3). An LMR > 3 proved to be prognostic in multivariate analysis (*p* = 0.019, HR 2.32), as did the presence of back pain (*p* = 0.014, HR 4.55). These were not significant in univariate analysis. For all patients and for the 110 patients that received neoadjuvant chemotherapy, the NLR was not a predictor of recurrence (*p* = 0.28, HR 1.34 and *p* = 0.90, HR = 0.96). However, for the 86 patients treated with upfront surgery, an NLR > 5 was associated with shorter recurrence-free survival (*p* = 0.026, HR = 2.5).

Tumour regression was assessed as a continuous variable (0–100%) in patients that underwent neoadjuvant chemotherapy. Higher residual tumour was associated with poorer prognosis in recurrence-free survival with an increased hazard of 1% for every 1% increase in residual tumour (*p* = 0.003, HR 1.01).

**Table 3 jcm-12-01989-t003:** Clinical and inflammatory prognostic markers of recurrence-free survival.

Recurrence-Free Survival				
	Univariate		Multivariate	
	*p*- Value	Hazard Ratio (95%CI)_	*p*- Value	Hazard Ratio (95% CI)
ECOG	0.228	1.27 (0.86–1.89)	0.825	1.1 (0.52–2.25)
CCI	0.981	1.00 (0.91–1.10)	0.566	0.95 (0.79–1.14)
Weight loss	0.346	1.20 (0.82–1.75)	0.935	1.03 (0.54–1.95)
Back pain	0.581	1.22 (0.60–2.51)	0.014 *	4.45 (1.34–14.26)
Surgical stage	0.001 *	1.36 (1.15–1.62)	0.003 *	1.66 (1.18–2.34)
NLR > 5	0.283	1.34 (0.79–2.27)	0.109	2.41 (0.82–7.04)
PLR > 150	0.671	1.08 (0.76–1.54)	0.975	1.00 (0.47–2.07)
LMR > 3	0.275	1.21 (0.86–1.71)	0.019 *	2.32 (1.15–4.68)
Bilirubin > 20 µmol/L	0.079	1.40 (0.96–2.04)	0.138	1.66 (0.85–3.26)
mGPS	0.238	0.82 (0.59–1.14))	0.459	0.84 (0.52–1.34)
CA 19–9 > 1000 kU/L	0.114	1.47 (0.91–2.38)	0.396	1.38 (0.65–2.92)

ECOG—Eastern Co-operative Oncology Group, CCI—Charleson comorbidity index, LMR—lymphocyte-to-monocyte ratio, PLR—platelet monocyte ratio, NLR—neutrophil monocyte ratio, mGPS—modified Glasgow performance scale. * Indicates statistical significance < 0.05.

#### 3.2.2. Overall Survival 

The neutrophil-to-lymphocyte ratio was a strong prognostic biomarker for overall survival. Patients with NLRs >5 had a median overall survival of only 13.0 months compared to 32.4 months in those ≤5 (Table 4, Figure 2). The NLR was statistically significant in both univariate (*p* = 0.001, HR 2.43) and multivariate analysis (*p* = 0.019, HR 3.04). Bilirubin levels were also a useful prognostic marker with levels above 20 µmol/L demonstrating significantly shorter overall survival at 26 months compared to those with <20 µmol/L at 42.7 months (*p* = 0.037, HR 1.52). Similar to recurrence-free survival, the surgical stage was also statistically significant, with the higher stages resulting in worse overall survival (*p* = 0.021, HR 1.23). Neither bilirubin nor surgical stage was significant on multivariate testing. Other inflammatory blood tests in the LMR and PLR were not prognostic for overall survival.

In addition, for patients diagnosed with recurrent disease, a higher NLR at diagnosis was a strong prognostic factor for shorter overall survival from the time of disease recurrence. Patients with a higher NLR (>5) at baseline had a median time from recurrence to overall survival of only 5 months versus 12.7 months in those with an NLR ≤ 5 (HR 2.37, 1.356–4.13, *p* = 0.002).

When divided by upfront treatment modality, an NLR > 5 was a poor prognostic marker in overall survival for both the neoadjuvant group (*p* = 0.023, HR 1.96) and the upfront surgery group (*p* = 0.001, HR 4.46).

Higher residual tumour was a prognostic marker for overall survival with an increased hazard of 1% for every 1% in remaining viable tumour (*p* = 0.023, HR 1.01).

**Table 4 jcm-12-01989-t004:** Clinical and inflammatory prognostic markers of overall survival.

Overall Survival				
	Univariate		Multivariate	
	*p*- Value	Hazard Ratio (95%CI)_	*p*- Value	Hazard Ratio (95% CI)
ECOG	0.144	1.30 (0.92–1.84)	0.785	1.10 (0.55–2.2)
CCI	0.383	1.04 (0.95–1.15)	0.583	1.05 (0.88–1.27)
Weight loss	0.147	1.33 (0.91–1.94)	0.964	0.99 (0.52–1.88)
Back pain	0.484	1.30 (0.63–2.66)	0.010 *	4.21 (1.4–12.63)
Surgical stage	0.021 *	1.23 (1.03–1.46)	0.092	1.33 (0.95–1.87)
NLR > 5	0.001 *	2.43 (1.54–3.85)	0.019 *	3.04 (1.20–7.71)
PLR > 150	0.163	1.30 (0.90–1.90)	0.504	1.33 (0.58–3.08)
LMR > 3	0.966	1.01 (0.70–1.45)	0.427	1.346 (0.65–2.80)
Bilirubin > 20 µmol/L	0.037 *	1.52 (1.03–2.26)	0.051	2.07 (1.00–4.27)
mGPS	0.234	0.81 (0.57–1.15)	0.028 *	0.58 (0.36–0.94)
CA 19–9 > 1000 kU/L	0.270	1.30 (0.81–2.16)	0.953	1.02 (0.46–2.30)

ECOG—Eastern Co-operative Oncology Group, CCI—Charleson comorbidity index, LMR—lymphocyte-to-monocyte ratio, PLR—platelet-to-monocyte ratio, NLR—neutrophil-to-monocyte ratio, mGPS—modified Glasgow performance scale. * Indicates statistical significance < 0.05.

## 4. Discussion

It is well recognised that the immune system and advanced cancer are closely linked, with many studies demonstrating the relationship between inflammatory blood tests and advanced pancreatic cancer [5,23]. What is less established is the role of these inflammatory markers in prognosis in early pancreatic cancer patients that undergo curative surgery, particularly in those that undergo neoadjuvant chemotherapy. In addition, there have been no studies assessing the role of these potential biomarkers in predicting chemotherapy response after neoadjuvant chemotherapy by comparing it to tumour regression in the surgical specimen. In our study, the estimated remaining viable tumour (as taken as a percentage estimate of the initial tumour) was used to assess the chemotherapy response. We used percentages as a continuous variable, rather than one specific scoring system as there is no consensus for which scoring system is the most accurate in pancreatic cancer. The purpose of our study was to establish the roles of these inflammatory markers as prognostic and predictive biomarkers in patients diagnosed with earlier disease that went on to have surgery.

We identified that the platelet-to-lymphocyte ratio (PLR) is a potential predictor for tumour viability, with a ratio > 150 associated with a higher residual viable tumour at the time of surgery. Our study also demonstrated that there is a role of neutrophil-to-lymphocyte ratio (NLR) in prognostication, with levels > 5 resulting in a significantly shorter median overall survival of 13 months compared to 32 months in those with a ratio of ≤5. This significant difference was not observed in either group in recurrence-free survival (RFS); however, in patients that underwent upfront resection, a higher NLR was associated with shorter RFS. 

This lack of difference in the recurrence-free survival for both groups may also tell us valuable information. It may be that the NLR is not as important early on in the disease course when other factors (including surgical outcomes) play an important role. 

In addition, for patients that did relapse, a higher NLR predicted for a shorter time to death (5 versus 12.7 months). This poorer survival on relapse may indicate that a higher baseline NLR is associated with more aggressive disease or may reflect patient factors including a poorer performance status.

This confirms our hypothesis that the NLR can be used as a surrogate prognostic marker for overall survival in patients that have an earlier stage of disease and undergo surgery. It also highlights the important role the immune system has in response to pancreatic cancer, which may prove fundamental in future therapies, particularly with the more widespread use of immunotherapy, a treatment that has so far proved disappointing in the treatment of this terrible disease.

The ability to identify prognostic and predictive markers is crucial to personalising treatment for patients in the clinic. Due to the toxicity of modern-day chemotherapy and the poor response rates of second- and third-line regimens, the ability to predict patients that will derive less benefit, particularly at the time of recurrence, may be useful in sparing them unnecessary side effects and preserving quality of life. 

There are limitations of these studies to note. Firstly, this is a retrospective review and there may be recall bias. In addition, white blood cells and c-reactive protein can be elevated for several reasons and pancreatic cancer can be linked to pancreatitis. We opted to use blood tests on admission to the hospital for staging laparoscopy to try to standardise the time of collection; however, this is a one-time point. Blood parameters are dynamic and may change over time. In addition, the correlation between a higher baseline PLR and residual tumour at the time of surgery, whilst statistically significant, has a coefficient of only 0.2 and hence should be interpreted with caution.

There are also strengths to our study. Although there have been publications involving the NLR and PLR in advanced and even resectable pancreatic cancer, this is the first paper, to our knowledge, that assesses the relationship between baseline inflammatory blood tests and tumour viability after neoadjuvant chemotherapy [9]. The ability to predict which patients may respond better to chemotherapy may be invaluable in determining the ideal initial treatment modality, a contentious issue in upfront resectable disease. A large number of patients (196) were included in the analysis and the blood tests were processed at one of only two laboratories at the same time in the patient’s clinical journey to minimise bias.

## 5. Conclusions

This real-world study demonstrates that there is an association between the immune system and earlier pancreatic cancer, with a higher neutrophil-lymphocyte ratio predicting shorter survival and a higher platelet-to-lymphocyte ratio correlating with greater tumour viability at the time of surgery in patients receiving neoadjuvant chemotherapy. Further prospective trials to validate these findings are crucial to provide insight into the role of inflammatory blood tests as potential biomarkers in early pancreatic cancer.

## Figures and Tables

**Figure 1 jcm-12-01989-f001:**
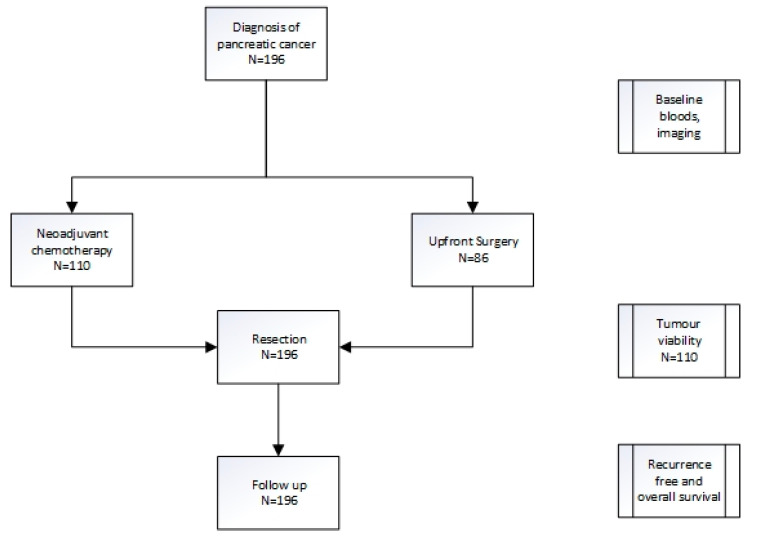
Treatment pathway for pancreatic cancer patients.

**Figure 2 jcm-12-01989-f002:**
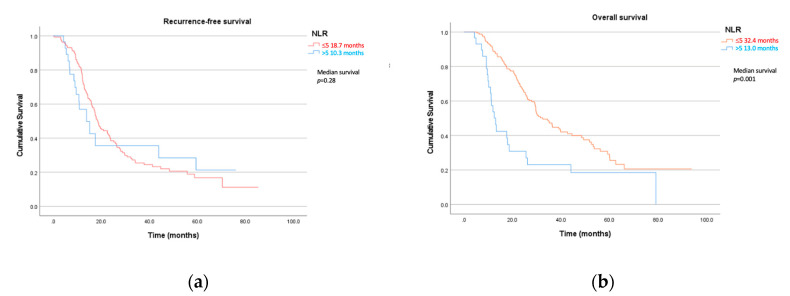
(**a**) Recurrence-free and (**b**) overall survival by neutrophil-to-lymphocyte ratio (NLR).

**Table 1 jcm-12-01989-t001:** Baseline characteristics.

Stage at Diagnosis (Radiological Stage)	Number of Patients
1	145
2	47
3	4
Resectability	
Upfront resectable	133
Borderline	26
Locally advanced	37
Upfront treatment	
Upfront resection	86
NAC	110
NAC regimen	
Gemcitabine plus nab-paclitaxel	41
FOLFIRINOX	60
Other	9
Charleson comorbidity	
<2	36
2–5	117
>5	41
Unknown	2
Sex	
M	104
F	92
Inflammatory markers	Median (range)
NLR	2.9 (0.2–17.6)
PLR	181 (12.6–662)
LMR	2.4 (0.4–18)

NAC—neoadjuvant chemotherapy, FOLFIRINOX—(5-fluorouracil, irinotecan, and oxaliplatin), NLR—neutrophil monocyte ratio, PLR—platelet monocyte ratio, LMR—lymphocyte-to-monocyte ratio.

**Table 2 jcm-12-01989-t002:** Clinical and inflammatory predictive markers correlation with tumour viability.

	*p*-Value	Correlation Coefficient (Spearmen’s)
ECOG	0.102	0.158
CCI	0.165	0.134
Weight Loss	0.997	0
Back pain	0.276	−0.106
NLR > 5	0.875	−0.015
PLR > 150	0.03 *	0.21
LMR > 3	0.075	−0.172
mGPS	0.436	−0.93
CA 19.9 > 1000 kU/L	0.818	0.023

ECOG—Eastern Co-operative Oncology Group, CCI—Charleson comorbidity index, NLR—neutrophil monocyte ratio, PLR—platelet monocyte ratio, LMR—lymphocyte-to-monocyte ratio, mGPS—modified Glasgow performance scale. * Indicates statistical significance < 0.05.

## Data Availability

Data are available in the paper.

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
