# Peer review of "The Prognostic and Predictive Role of the Neutrophil-to-Lymphocyte Ratio (NLR), Platelet-to-Lymphocyte Ratio (PLR), and Lymphocyte-to-Monocyte Ratio (LMR) as Biomarkers in Resected Pancreatic Cancer"

_jcm, 2023, doi:10.3390/jcm12051989_

Round 1

Reviewer 1 Report

Author investigated prognostic and predictive for tumor viability of pre-NACT (neoadjuvant chemotherapy) inflammatory markers including neutrophil-lymphocyte ratio (NLR), platelet lymphocyte ratio (PLR), lymphocyte monocyte ratio (LMR). They found PLR was predictive marker for tumor viability, and NLR was prognostic indicator for survival in this study. However, there are some discussion points in this study as follows:

1.     What is the definition of tumor viability? Although author followed AJCC 8th edition, further explanation is needed for readers. The most common used criteria is CAP classification, However, there is a issue for interobserver disagreement and this classification has a limitation for stratification of survival. The patient experiences early recurrence and shorter survival period even in CAP 1 regressed tumor. Furthermore, patients without tumor viability after neoadjuvant chemotherapy are low proportion. Author had better the proportion of tumor viability with survival analysis in this cohort and summarize current literatures of tumor viability in discussion.

2.     NLR was revealed as prognostic factor for survival, however, it showed no relationship in recurrence. NLR may be not prognostic marker for strict definition. Author had better describe this limitation in discussion.

3.     Timing of blood test: author used blood sample in pre-NACT status, this study is meaningful for selection of NACT before chemotherapy. However, author did not describe of the change of NLR, PLR, or LMR after chemotherapy. Regressed or aggravated pattern of inflammatory markers may be prognostic.

4.     Furthermore, author did not show perioperative outcomes according to NLR, PLR, and LMR. Author describe lower NLR patients showed lower stage than higher NLR patients. Author had better further describe this difference.  

5.     Finally, author commented different inflammatory tumor microenvironment in introduction. Author had better show pathologic difference according to the inflammatory markers with representative features.

Author Response

Thank you for taking the time to review our manuscript titled ‘The prognostic and predictive role of neutrophil lymphocyte ratio (NLR), platelet lymphocyte ratio (PLR), and lymphocyte monocyte ratio (LMR) as biomarkers in resected pancreatic cancer’.

Please find attached comments with references to changes in the text.

Review 1

Author investigated prognostic and predictive for tumor viability of pre-NACT (neoadjuvant chemotherapy) inflammatory markers including neutrophil-lymphocyte ratio (NLR), platelet lymphocyte ratio (PLR), lymphocyte monocyte ratio (LMR). They found PLR was predictive marker for tumor viability, and NLR was prognostic indicator for survival in this study. However, there are some discussion points in this study as follows:

  1. What is the definition of tumor viability? Although author followed AJCC 8thedition, further explanation is needed for readers. The most common used criteria is CAP classification, However, there is a issue for interobserver disagreement and this classification has a limitation for stratification of survival. The patient experiences early recurrence and shorter survival period even in CAP 1 regressed tumor. Furthermore, patients without tumor viability after neoadjuvant chemotherapy are low proportion. Author had better the proportion of tumor viability with survival analysis in this cohort and summarize current literatures of tumor viability in discussion.

The term tumour viability has been replaced by tumour regression in the text. A further explanation of this term is given in:

Background: lines 89-97

Tumour regression is assessed in the histopathological specimen in patients that have undergone neoadjuvant chemotherapy. It is an assessment of the proportion of viable tumour cells remaining in the surgical specimen in relation to the initial tumour volume15. As the initial volume is unknown it is an estimate and takes into consideration the degree of fibrosis post neoadjuvant treatment 16. At present there are many tumour regression scores that are used, the most common being the college of American pathologists (CAP) score and Evans criteria, however, there exists significant interobserver variability using these tools17. Moreover, while it allows for an assessment of the chemosensitivity of the tumour, its use as a prognostic marker has not been established.

Methods: lines 179-182

Tumour regression was assessed only on histopathological specimens of patients that received NAC. It was a marker of viable tumour remaining based of an estimate of initial tumour volume. This was assessed as a percentage and measured as a continuous variable. No scoring system was used due to the lack of consensus on tumour regression grading system in pancreatic cancer.

Discussion: lines 311-317

In addition, there have been no studies assessing the role of these potential biomarkers to predict chemotherapy response after neoadjuvant chemotherapy by comparing to tumour regression in the surgical specimen. In our study the estimated remaining viable tumour (as taken as a percentage estimate of initial tumour) was used to assess the chemotherapy response. We used percentages as a continuous variable, rather than one specific scoring system as there is no consensus for which scoring system is the most accurate in pancreatic cancer.

In addition, the impact that tumour regression had on survival is now addressed in the

Results sections line 255-258 and 287-289

Tumour regression was assessed as a continuous variable (0-100%) in patients that underwent neoadjuvant chemotherapy. Higher residual tumour was associated with poorer prognosis in recurrence free survival with an increase hazard of 1% for every 1% increase in residual tumour (p=0.003, HR 1.01).

Higher residual tumour was a prognostic marker for overall survival with increase hazard of 1% for every 1% in remaining viable tumour (p=0.023, HR 1.01).

  1. NLR was revealed as prognostic factor for survival, however, it showed no relationship in recurrence. NLR may be not prognostic marker for strict definition. Author had better describe this limitation in discussion.

To further identify the relationship between NLR and recurrence free survival the two groups were analysed separately depending on upfront treatment. For patients with upfront resection NLR did predict for recurrent free survival, however this was not evident in the neoadjuvant group.

A line has been added into the results for this

Results line: 250-254

For all patients and for the 110 patients that received neoadjuvant chemotherapy NLR was not a predictor of recurrence (p=0.28, HR 1.34 and p=0.90, HR=0.96). However, for the 86 patients treated with upfront surgery NLR>5 was associated with a shorter recurrence free survival (p=0.026, HR=2.5)

This has also been addressed in the discussion: Lines 324-333

This significant difference was not observed in both groups in recurrence free survival (RFS), however, in patients that underwent upfront resection, higher NLR was associated with shorter RFS.

This lack of difference in the recurrence free survival for both groups may also tell us valuable information. It may be that NLR in not as important early on in disease course when other factors (including surgical outcomes) play an important role.

In addition, for patients that did relapse, higher NLR predicted for shorter time to death (5 versus 12.7 months). This poorer survival on relapse may indicate that higher baseline NLR is associated with a more aggressive disease or may reflect patient factors including a poorer performance status.

  1. 3.     Timing of blood test: author used blood sample in pre-NACT status, this study is meaningful for selection of NACT before chemotherapy. However, author did not describe of the change of NLR, PLR, or LMR after chemotherapy. Regressed or aggravated pattern of inflammatory markers may be prognostic.

Only the blood tests from the admission for staging laparoscopy before neoadjuvant chemotherapy were used as chemotherapy and granulocyte colony stimulating factor would have a significant impact on the neutrophils, lymphocytes and platelets, thus making them uninterpretable. The aim of this study was to assess if baseline blood tests (before intervention) were useful predictive or prognostic biomarkers.  

4a).     Furthermore, author did not show perioperative outcomes according to NLR, PLR, and LMR.

  1. b) Author describe lower NLR patients showed lower stage than higher NLR patients. Author had better further describe this difference.  

  1. The aim of this study was not to assess neutrophil lymphocyte ratio (NLR), platelet lymphocyte ratio (PLR) and lymphocyte monocyte ratio (LMR) in relation to perioperative outcomes. Mortality after surgery was not a common outcome in our cohort with nil deaths within one hundred twenty days of surgery.
  2. The relationship between lower NLR and lower stage was not analysed in this study. I have deleted line 263, hopefully this clarifies for the reader/reviewer.

  1. Finally, author commented different inflammatory tumor microenvironment in introduction. Author had better show pathologic difference according to the inflammatory markers with representative features.

Different tumour microenvironments were not assessed in this paper. This has been removed from the introduction to avoid confusion.

Reviewer 2 Report

Abstract: Suggest replacing "in inflammatory blood tests as potential biomarkers" with "potential inflammatory biomarkers". I suggest authors clarify " higher neutrophil lymphocyte ratio". It is unclear if this indicates peripheral blood or tissue samples. The same applies to "platelet lymphocyte ratio".

Please clarify this statement:  "increased tumour viability in the surgical specimen in patients receiving neoadjuvant chemotherapy". Are the authors assessing in-vitro (tumor survival) or survival of patients. 

Introduction: May consider a concise introduction.

Methods:

Serum albumin, CRP, neutrophil count, and lymphocyte count can vary significantly from day to day. Did authors have a specified or standardized time for blood collection? Since this is a retrospective study the lab work must have already been done and there may be multiple lab results. How did the authors determine which one to select? There are multiple confounders to the blood cell counts and can represent acute inflammation of any source, medication effect, physiologic or pathologic stress or an unidentified infection.

Results: Most patients, 145/196 had stage I tumor, and this represents a highly biased sample of pancreatic cancer by any means. Patient outcome is expected to be better in this cohort of patients with localized pancreatic cancer. However, this does not represent the pancreatic patient population at large.

PLR was shown to predict tumor viability while NLR did not affect tumor viability while it predicted OS. Furthermore, in multivariate analysis tumor stage did not predict OS while NLR >5 predicted poor OS. This finding is questionable, as we know that tumor stage is the most important predictor of pancreatic cancer outcome and OS. These results are interesting and have to be interpreted with caution.

Author Response

Thank you for taking the time to review our manuscript titled ‘The prognostic and predictive role of neutrophil lymphocyte ratio (NLR), platelet lymphocyte ratio (PLR), and lymphocyte monocyte ratio (LMR) as biomarkers in resected pancreatic cancer’.

Please find attached comments with references to changes in the text.

Review 2

Abstract: Suggest replacing "in inflammatory blood tests as potential biomarkers" with "potential inflammatory biomarkers". I suggest authors clarify " higher neutrophil lymphocyte ratio". It is unclear if this indicates peripheral blood or tissue samples. The same applies to "platelet lymphocyte ratio".

This has been changed. Please see line 20.

In peripheral blood was added. Please see line 22.

Please clarify this statement:  "increased tumour viability in the surgical specimen in patients receiving neoadjuvant chemotherapy". Are the authors assessing in-vitro (tumor survival) or survival of patients. 

The term ‘surgical specimen’ was changed to ‘histopathological specimen’ line 28. Tumour viability was changed to ‘residual tumour’. The predictive factors were compared to the tumour regression in the resected tumour tissue at time of surgery. The prognostic factors these were compared to both recurrence free and overall survival of the patient.

Introduction: May consider a concise introduction.

Methods:

Serum albumin, CRP, neutrophil count, and lymphocyte count can vary significantly from day to day. Did authors have a specified or standardized time for blood collection? Since this is a retrospective study the lab work must have already been done and there may be multiple lab results. How did the authors determine which one to select? There are multiple confounders to the blood cell counts and can represent acute inflammation of any source, medication effect, physiologic or pathologic stress or an unidentified infection.

We acknowledge that patients get numerous blood tests throughout their work up. As such, prior to commencement of this study the authors chose the one time point to analyse patient bloods. This was done to standardise the results. All bloods were taken at point of admission (before) the patient’s staging laparoscopy. Repeat blood tests after chemotherapy were not assessed as chemotherapy and colony growth stimulating factor would have significantly impacted on results.

Results: Most patients, 145/196 had stage I tumor, and this represents a highly biased sample of pancreatic cancer by any means. Patient outcome is expected to be better in this cohort of patients with localized pancreatic cancer. However, this does not represent the pancreatic patient population at large.

This was poorly worded and has been corrected in table 1. This was their upfront radiological stage. There were however, a significant proportion of patients with earlier stage as noted by their resectability status (133/196). This was intentional to choose patients with earlier disease, as the role of NLR in patients with more advanced pancreatic cancer has been more established.

PLR was shown to predict tumor viability while NLR did not affect tumor viability while it predicted OS. Furthermore, in multivariate analysis tumor stage did not predict OS while NLR >5 predicted poor OS. This finding is questionable, as we know that tumor stage is the most important predictor of pancreatic cancer outcome and OS. These results are interesting and have to be interpreted with caution.

Whilst the surgical stage was not significant in multivariate testing, it was significant in univariate testing. This may have to do with the impact of the other variables in the multivariate analysis.